# The life cycle of *Drosophila* orphan genes

**Nicola Palmieri, Carolin Kosiol, Christian Schlötterer***

Institut für Populationsgenetik, Vetmeduni Vienna, Vienna, Austria

**Abstract** Orphans are genes restricted to a single phylogenetic lineage and emerge at high rates. While this predicts an accumulation of genes, the gene number has remained remarkably constant through evolution. This paradox has not yet been resolved. Because orphan genes have been mainly analyzed over long evolutionary time scales, orphan loss has remained unexplored. Here we study the patterns of orphan turnover among close relatives in the *Drosophila obscura* group. We show that orphans are not only emerging at a high rate, but that they are also rapidly lost. Interestingly, recently emerged orphans are more likely to be lost than older ones. Furthermore, highly expressed orphans with a strong male-bias are more likely to be retained. Since both lost and retained orphans show similar evolutionary signatures of functional conservation, we propose that orphan loss is not driven by high rates of sequence evolution, but reflects lineage-specific functional requirements.

## Introduction

Orphans are genes with limited phylogenetic distribution and represent a considerable fraction (up to 30%) of the gene catalog in all sequenced genomes (*Khalturin et al., 2009*). Studies conducted in different eukaryotes showed that orphans emerge at high rates (*Domazet-Loso et al., 2007*; *Wissler et al., 2013*). While gene duplication and exaptation from transposable elements often result in orphan genes (*Toll-Riera et al., 2009*), they also originate frequently de novo from non-coding DNA (*Cai et al., 2008*; *Heinen et al., 2009*; *Knowles and McLysaght, 2009*; *Wu et al., 2011*; *Yang and Huang, 2011*; *Xie et al., 2012*; *Neme and Tautz, 2013*; *Wu and Zhang, 2013*), probably through intermediate proto-genes (*Carvunis et al., 2012*). Compared to evolutionary conserved genes, orphans are overall shorter (*Lipman et al., 2002*), fast evolving (*Domazet-Loso and Tautz, 2003*), have lower and more tissue-restricted expression (*Lemos et al., 2005*). Moreover, they often show testis-biased expression (*Levine et al., 2006*; *Begun et al., 2007*), probably due to frequent origination in testis (*Kaessmann, 2010*).

In *Drosophila* the rate of orphan emergence is particulary high (*Domazet-Loso et al., 2007*) and many orphans become quickly essential (*Chen et al., 2010*). Although the function of only a few orphan genes has been studied, it has been proposed that orphans might serve an important role in speciation and adaptation to different environments (*Khalturin et al., 2008*; *Khalturin et al., 2009*; *Colbourne et al., 2011*). The high rate of orphan origination would predict an increase in gene content over time. However, gene content in eukaryotes is remarkably stable compared to genome size, as highlighted by *Tautz and Domazet-Loso (2011)*. To solve this paradox Tautz and Domazet-Loso proposed that orphans have only a short lifetime ('rapid-turnover' hypothesis) (*Tautz and Domazet-Loso, 2011*). Thus, although orphans are continuously created, most of them might be lost in a relatively short evolutionary time. Relaxed selective costraints in orphans (*Cai and Petrov, 2010*) might also contribute to the high rate of orphan loss.

Moreover, since orphans are typically identifed by the comparison of distantly related species, their evolutionary stability has been so far neglected. This contrasts the comprehensive analysis of

*For correspondence: christian.
schloetterer@vetmeduni.ac.at

**Competing interests:** The authors declare that no competing interests exist.

**Reviewing editor**: Diethard Tautz, Max Planck Institute for Evolutionary Biology, Germany

**eLife digest** New genes are added to most genomes on a steady basis. A new gene can either begin as a copy of an existing gene from elsewhere in the genome, or is created entirely 'from scratch' from a DNA sequence that had not previously encoded for a protein. New genes that are not found in other related species are called orphan genes—and these genes can account for up to 30% of all the genes in the well-studied genomes. However, for reasons that are not fully understood, the total number of genes in most genomes remains fairly constant despite these regular additions. Now, Palmieri et al. have investigated this paradox by following the evolutionary fate of orphan genes in a small group of related species of fruit fly.

Palmieri et al. discovered that most orphan genes are very short-lived, even though they showed clear signals of carrying out important functions. Most orphan genes died out quickly due to mutations that made them unable to be expressed as functional proteins, and a small number were deleted entirely from the genome. Unexpectedly, new orphan genes were more likely to die out than those that had been around for a while.

Palmieri et al. also found that the expression levels of orphan genes determined their probability of dying with those genes that were expressed to the highest levels being most likely to persist longer. Furthermore, genes that were expressed more in males than in females were also less likely to die. The next challenge will be to identify the mechanisms that determine which orphan genes survive and which do not.

evolutionary patterns of gains and losses of non-orphan genes (*Hahn et al., 2007*). In this study, several partially interrelated factors affect gene loss, including gene expression levels, number of protein–protein interactions, gene dispensability, and rate of sequence evolution (*Krylov et al., 2003*; *Cai and Petrov, 2010*).

This study focuses for the first time on the evolutionary stability of orphan genes. We investigate the factors contributing to orphan loss and find that orphan age, male-biased gene expression, and microsatellite content are correlated with orphan stability. Surprisingly, differences in evolutionary rates cannot explain orphan loss and we propose that orphan loss is driven by lineage-specific evolutionary constraints. Overall, orphan genes are lost at a significantly higher rate than non-orphan genes, supporting the 'rapid-turnover' hypothesis.

## Results

Orphans are commonly detected by BLASTing the genes of a given organism against a set of outgroup species (*Domazet-Loso and Tautz, 2003*; *Toll-Riera et al., 2009*). A BLASTP cutoff of $10^{-3}$–$10^{-4}$ was found to be optimal to maximize sensitivity and specificity in *Drosophila* (*Domazet-Loso and Tautz, 2003*). To identify orphans we used a BLASTP cutoff of $10^{-4}$ combined with a TBLASTN cutoff of $10^{-4}$, to exclude genes with unannotated orthologs in other species. Following these criteria, we searched in *Drosophila pseudoobscura* for genes with no sequence conservation in 10 *Drosophila* species outside the *Drosophila obscura* group (*Figure 6—figure supplement 1*). In total, we identified 1152 orphans, corresponding to 7% of all the *D. pseudoobscura* genes. Our estimate is slightly lower than a previous one (*Zhang et al., 2010*), due to our different filtering procedure, but still consistent with a high rate of orphan gain in *Drosophila* (*Domazet-Loso and Tautz, 2003*; *Domazet-Loso et al., 2007*; *Zhou et al., 2008*; *Wissler et al., 2013*). Our data clearly indicate that orphan genes are subject to purifying selection, as they show several hallmarks of functional protein-coding sequences (*Figures 1, 2*). A comparison of orphan genes preserved between *D. pseudoobscura* and *D. affinis* resulted in a distribution of *dN/dS* significantly lower than 1 with a median of 0.44 (*Figure 1—figure supplement 1*, one-sided Wilcoxon signed-rank test, $p<1.0 \times 10^{-15}$), as expected for protein-coding sequences. Moreover, *dN/dS* for orphans is significantly lower (Mann–Whitney test, $p=2.7 \times 10^{-14}$) than *dN/dS* calculated on a random set of intergenic regions with the same length distribution of orphans (see 'Materials and methods', section 'Evolutionary rates') (*Figure 1A*). Consistent with this, we also found orphans to be more

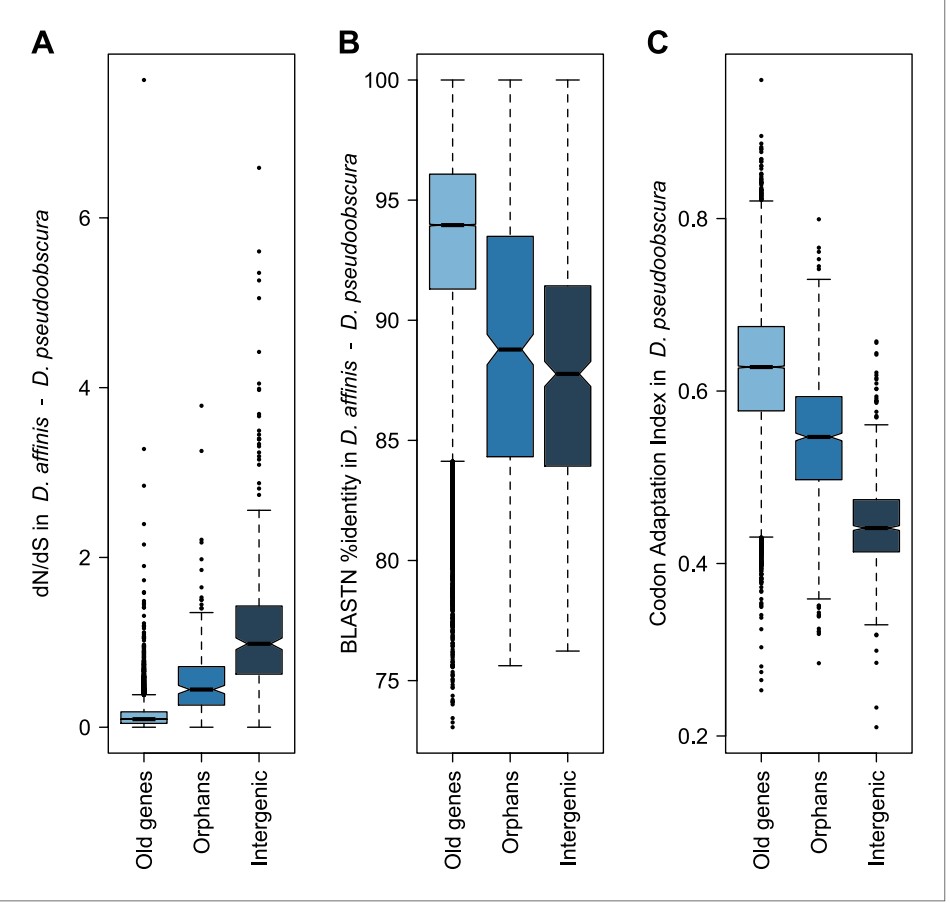

**Figure 1**. Orphans are subject to purifying selection. (**A**) *dN/dS* of *D. pseudoobscura* and *D. affinis* orthologs. *dN/dS* is lowest for old genes, but also orphan genes have *dN/dS* smaller than one. A comparison of orphans and intergenic regions shows that *dN/dS* for orphans is significantly smaller (Mann–Whitney test, $p=9.5 \times 10^{-10}$), indicating purifying selection on orphan genes. Intergenic regions were of similar length and chromosomal position as the orphan genes. (**B**) Sequence similarity in HSPs obtained from BLASTing *D. pseudoobscura* genes against the *D. affinis* genome. Orphans are more conserved than intergenic regions (Mann–Whitney test, $p=0.00238$) and less conserved than old genes (Mann–Whitney test, $p<1.0 \times 10^{-15}$). (**C**) Codon usage was measured by the Codon Adaptation Index (***Sharp and Li, 1987***). The codon usage of orphans is significantly higher than that of intergenic regions (Mann–Whitney test, $p<1.0 \times 10^{-15}$) indicating that orphans are subject to purifying selection. In comparison to old genes, orphans have a significantly lower codon usage bias (Mann–Whitney test–$p<1.0 \times 10^{-15}$). Overall, all three analyses demonstrate that orphans are not annotation artifacts, but evolutionary conserved genes.

The following figure supplements are available for figure 1:

**Figure supplement 1**. Distribution of *dN/dS* for orphan genes.

**Figure supplement 2**. Conservation of orphans in the obscura group.

conserved than intergenic regions (***Figure 1B***, ***Figure 1—figure supplement 2***). The codon usage bias of orphans is intermediate to that of old genes and intergenic regions (***Figure 1C***).

To further test for purifying selection acting on orphans, we used a polymorphism dataset of 45 strains being re-sequenced for the third chromosome of *D. pseudoobscura* ('Materials and methods'). We calculated the ratio of synonymous to non-synonymous polymorphism (*pN/pS*), since it provides an indication of purifying selection. We found that *pN/pS* for orphans is significantly lower compared to intergenic regions (Mann–Whitney test, $p=0.02182$) (***Figure 2***), and significantly greater for old genes (Mann–Whitney test, $p<1.0 \times 10^{-15}$), consistent with purifying selection operating on orphans.

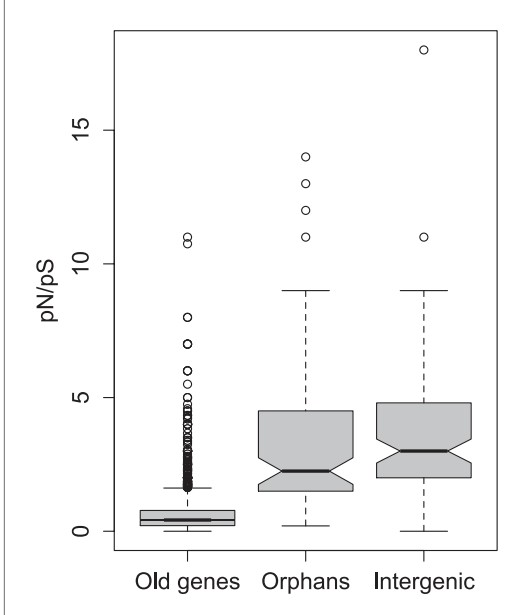

**Figure 2**. pN/pS for old genes, orphans, and intergenic regions. Orphans show a *pN/pS* intermediate between old genes and intergenic regions. Nevertheless, *pN/pS* is significantly smaller for orphans compared to intergenic regions (Mann–Whitney test, p<1.0 × 10⁻¹⁵), indicating coding purifying selection acting on orphans.

In agreement with studies in other species (*Domazet-Loso and Tautz, 2003*; *Toll-Riera et al., 2009*; *Wolf et al., 2009*; *Cai and Petrov, 2010*; *Capra et al., 2010*; *Carvunis et al., 2012*), we also find that orphan genes are shorter (median length for orphans = 344 bp, median length for old genes = 1470 bp), have a lower GC content (median GC content for orphans = 0.54, median GC content for old genes = 0.55), are expressed at lower levels (expression in *D. pseudoobscura* males: mean expression for orphans = 29 FPKM, mean expression for old genes = 41 FPKM) than old genes (*Figure 3*). Using CD-Hit (*Li and Godzik, 2006*), we found the fraction of genes with a paralog (>90% protein similarity) to be similar for orphans (6.9%) and old genes (6.4%). Orphans are more enriched in microsatellites, also consistent with previous findings in vertebrates (*Toll-Riera et al., 2012*) and rice (*Guo et al., 2007*). Furthermore, unlike mammals (*Toll-Riera et al., 2009*), none of the *D. pseudoobscura* orphans was found to be associated with transposable elements (see 'Transposons detection').

The distribution of orphans is heterogeneous across chromosomes ($\chi^2$-test, p<1.0 × 10⁻¹⁵), with the X chromosome having the highest fraction of orphans. In the *obscura* group, the two X-chromosome arms have a different evolutionary history. XL corresponds to Muller's element A and is homologous to the X chromosome in *D. melanogaster*. XR, however, has been recently derived from an autosome (Muller's element D, 3L in *D. melanogaster*). Analyzing the old-X and neo-X chromosomes separately, we observed a striking difference in the number of orphans despite similar chromosome sizes, with the old-X responsible for the excess of X-linked orphan genes, and the neo-X showing a similar number of orphans as the autosomes (*Figure 4*). For each chromosomal arm, we computed genomic features in 100 kb windows to correlate them with the difference in orphan content between old-X and neo-X. We found that average GC content, microsatellite density, transposon density, and length of intergenic regions differ between the two chromosomal arms (*Figure 5*).

We hypothesized that this pronounced difference between the two chromosome arms might reflect a different history of X-linkage. If orphan genes emerge at a higher rate on the X-chromosome (*Levine et al., 2006*), the shorter history of X-linkage on the neo-X could explain the paucity of orphans on the neo-X compared to old-X. In this case, the difference in orphan number between old-X and neo-X chromosomes should date back to the time before the origin of the neo-X, with a similar number of orphans originating after the creation of the neo-X. We therefore used the genomic sequences of five members of the *D. obscura* group (*D. pseudoobscura* [*Richards et al., 2005*], *D. miranda* [*Zhou and Bachtrog, 2012*], and the de novo assembled *D. persimilis*, *D. lowei*, and *D. affinis*) to date the origin of the orphan genes to different ancestral nodes in the phylogenetic tree of these species (*Beckenbach et al., 1993*). We distinguished five groups of genes: old genes (non orphans) and four different orphan age classes (*Figure 6*). Surprisingly, we observed a consistent paucity of orphans on XR relative to XL across all age classes (*Figure 7*). This persistent difference in orphan number between XL and XR in all age classes suggests that X-linkage is not sufficient to explain the enrichment of orphans on XL. We conclude that the former autosome differs from the ancestral X chromosomal arm by a yet unidentified feature that affects the emergence of new orphans.

The analysis of orphans that have putatively lost their function via the acquisition of a stop codon or a frame shift causing insertion/deletion (pseudogenized/lost orphans) reveals another interesting feature of the XL–XR fusion. The oldest orphans in our dataset (age class 4) show a

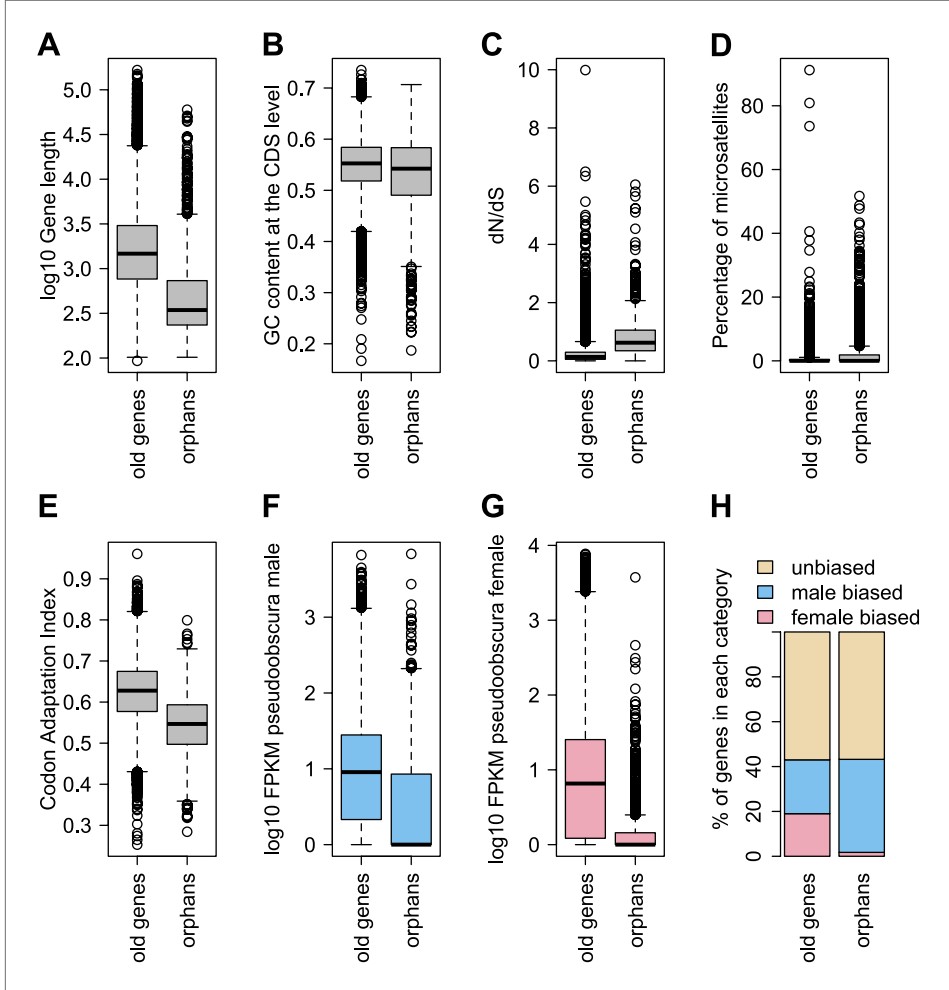

**Figure 3**. Comparison of orphans and genes conserved among 10 Drosophila species outside of the *obscura* group. Orphans differ from old genes in various features: (**A**) gene length (**B**) GC content, (**C**) *dN/dS* (**D**) percentage of microsatellites in coding sequence (**E**) Codon Adaptation Index (**F**) gene expression level in *D. pseudoobscura* males (**G**) gene expression level in *D. pseudoobscura* females (**H**) sex-biased expression. Orphans are shorter (Mann–Whitney test, $p < 1.0 \times 10^{-15}$), have lower GC content (Mann–Whitney test, $p = 3.9 \times 10^{-7}$), lower codon usage bias (Mann–Whitney test, $p < 1.0 \times 10^{-15}$), lower expression (Mann–Whitney test, $p < 1.0 \times 10^{-15}$), higher proportion of microsatellites (Mann–Whitney test, $p = 1.8 \times 10^{-4}$) and higher *dN/dS* (Mann–Whitney test, $p < 1.0 \times 10^{-15}$) compared to old genes. Moreover, orphans are more enriched in male-biased genes compared to old genes ($\chi^2$-test, $p < 1.0 \times 10^{-15}$).

pronounced excess of pseudogenized orphans on XR in *D. affinis* and *D. miranda* (*Figure 8A*). This trend was not observed for orphans that emerged on XR after the XL–XR fusion (*Figure 8B,C*), nor for old genes (*Figure 8D*) and is not due to an increased rate of orphan gain on XR (*Figure 9*). Since the oldest orphans (age class 4) on XR are a mixture of autosomal (i.e., before the fusion) and sex-chromosomal (i.e., after the fusion) orphans, we speculate that the high rate of pseudogenization of orphans on the XR may reflect the new X-linkage of previously autosomal orphans. A previous study (*Meisel et al., 2009*) found that the XR chromosome has experienced a burst of gene duplications to autosomes after its creation. It is plausible that after the conversion of the XR from autosome to sex-chromosome, orphans might have been duplicated to autosomes, whereas the XR ancestral copy would have become pseudogenized. To test this hypothesis, we looked for evidence of gene duplications for the orphans lost on the XR at node 4 (*Figure 6*). We aligned the sequences of these genes in *D. lowei* and *D. miranda* to the respective genomes using BLASTN (cutoff $10^{-5}$). Upon manual inspection of the alignments, we found that only 1 out of 21 genes in *D. miranda*

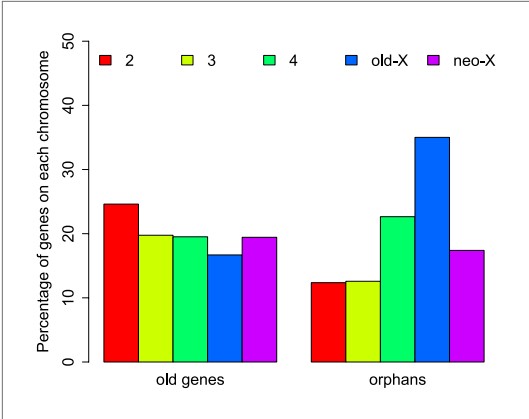

**Figure 4**. Chromosomal distribution of old genes and orphan genes. Orphans are overrepresented on the old-X. The number of orphan genes on the neo-X (XR) is significantly lower than on the old-X (XL) ($\chi^2$-test, p<1.0 × 10$^{-15}$).

(gene ID: GA23486) and 1 out of 14 genes in *D. lowei* (gene ID: GA23807) had a second hit on an autosome covering at least 50% of the length of the query gene. Other genes either produced a single best hit on the XR chromosome or spurious short hits on other chromosomes (data not shown). Thus, we conclude that duplication of orphans cannot explain the excess of pseudogenized orphans on XR. Nevertheless, our analysis clearly indicates that the emergence of the neo-X chromosome influenced the orphan dynamics on XR, affecting rates of both gain and loss, thus we excluded this chromosome arm for our analyses of the rate of orphan turnover.

For each age class, we determined the number of pseudogenized orphans (*Tautz and Domazet-Loso, 2011*). In the *D. persimilis* lineage, orphan pseudogenization can be studied for three different age classes. If orphans of all age classes were functionally equivalent, no difference in the rate of orphan pseudogenization would be expected. We observe, however, that the fraction of orphan pseudogenes decreases with age (*Figure 10*). The *D. miranda* lineage also shows a higher loss of young orphan genes. The relatively small number of observations, however, precludes statistical testing of this trend. Overall, orphan genes are lost significantly more often than old genes (Fisher's exact test, p=3.3 × 10$^{-8}$), consistent with the rapid turnover hypothesis. The unequal conservation of orphans of different age classes is also apparent after normalizing by coding sequence length (*Figure 11*), to account for the fact that longer coding sequences (CDS) have a greater chance of acquiring ORF-disrupting mutations. When looking at the distribution of premature termination codons (PTC) along the open reading frame (ORF) of all genes, we observed that PTCs are enriched at the beginning and at the end of the ORF (*Figure 12*), consistent with previous results in *D. melanogaster* (*Lee and Reinhardt, 2012*) and *D. pseudoobscura* (*Hoehn et al., 2012*). Since ORF-disrupting mutations occuring at the end of the ORF might have little impact on gene function, we redefined pseudogenes by considering only ORF-disrupting mutations localized in the first half of the ORF and confirmed that orphans of age class 3 are lost more often than those of age class 4 (*Figure 13*). Age class 2 was intermediate, most likely not reflecting a biological phenomenon, but due to a high sampling variance associated with the small number of observations (9 orphans). Finally, the pattern is also robust to a more conservative criterion for ortholog assignment (see 'Annotation of the *obscura* species', *Figure 14*).

To determine features associated with the differences in disabling mutations among orphans from different age classes, we contrasted orphans lost in *D. lowei* and/or *D. persimilis* (lost orphans) vs orphans conserved in all the *obscura* species (conserved orphans). Genes in both classes evolve at the same rate, are of similar length, and have similar codon usage bias (*Figure 15A–E*). Conserved orphans have a higher GC content, contain fewer microsatellites, are expressed at a higher level and are more male-biased (*Figure 15B,D–F,G,H*) compared to lost orphans. Conserved orphans tend to increase their expression level as they become older (*Figure 16A*), whereas the opposite pattern is true for lost orphans (*Figure 16B*).

Orphan genes are frequently expressed in the testis (*Levine et al., 2006*; *Begun et al., 2007*) and have a male-biased gene expression pattern (*Metta and Schlötterer, 2008*). This pattern could be generated by pervasive gene expression in testis, which facilitates the functional recruitment of non-specific expression (*Kaessmann, 2010*). Another explanation is that expression in testis does not require a complex architecture of regulatory modules (*Sassone-Corsi, 2002*; *Kleene, 2005*; *Kaessmann, 2010*), so that fewer substitutions are required to obtain a functional regulatory module for expressing a novel gene in testis compared to other tissues. We scrutinized these explanations by comparing the fraction of male-biased genes among orphan genes from different age classes. Unexpectedly, the fraction of male-biased genes increases with the age of the orphan genes

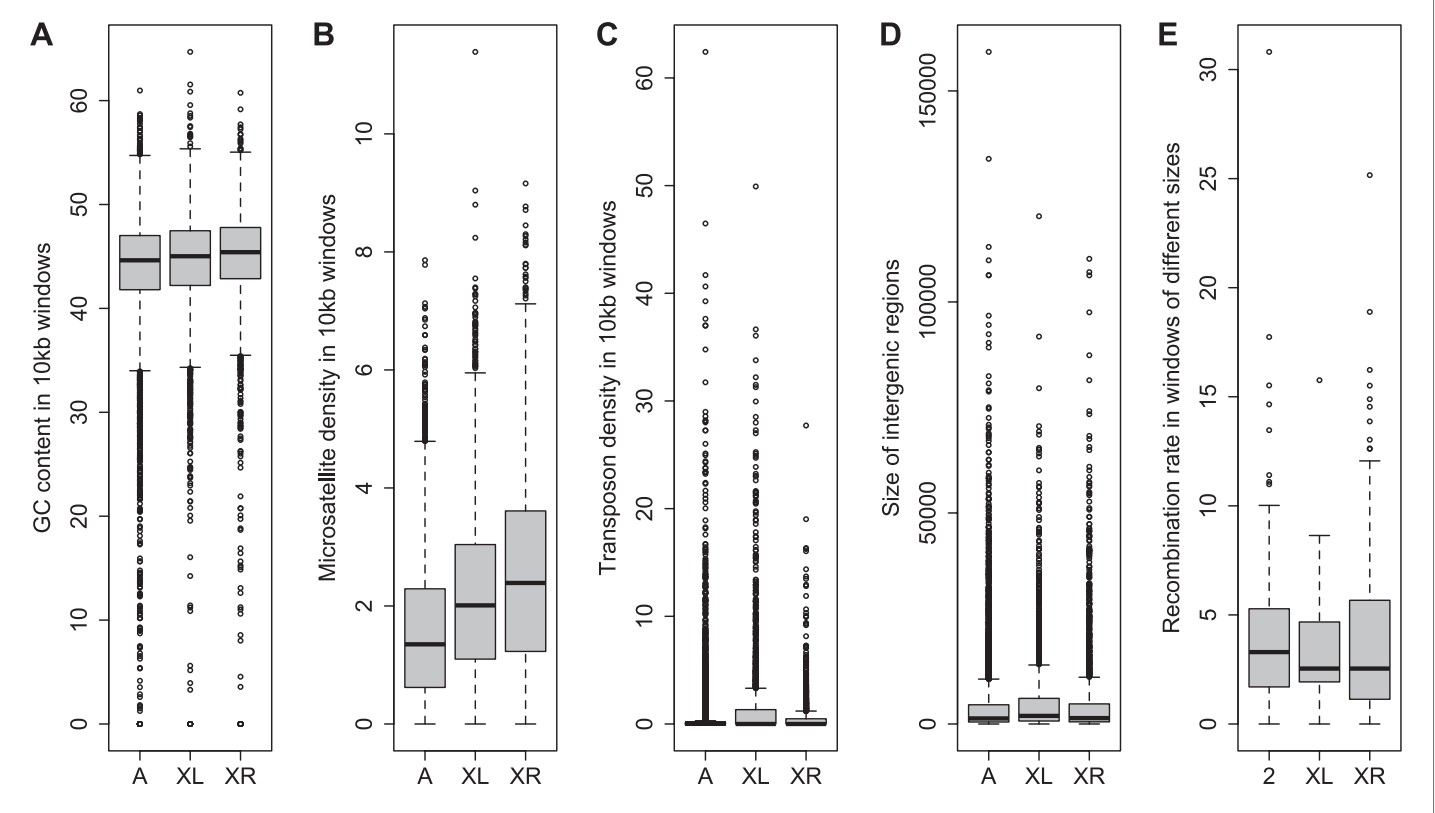

**Figure 5**. Comparison of genomic features among autosomes, old-X and neo-X. (**A**) GC content in 100 kb windows, (**B**) Microsatellite density in 100 kb windows, (**C**) Transposon density in 100-kb windows, (**D**) Length of intergenic regions, (**E**) Recombination rate. GC content is significantly greater on the neo-X compared to old-X for 10 kb windows (Mann–Whitney test, p=0.00020), but not for 100 kb windows (Mann–Whitney test, p=0.1092). Microsatellite density is significantly higher on the neo-X for both windows of 10 kb (Mann–Whitney test, p=$1.9 \times 10^{-12}$) and 100 kb (Mann–Whitney test, p=0.00025). Transposon density is significantly lower on the neo-X for both windows of 10 kb (Mann–Whitney test, p<$1.0 \times 10^{-15}$) and 100 kb (Mann–Whitney test, p=$4.6 \times 10^{-12}$). Intergenic regions are significantly shorter on the neo-X compared to the old-X (Mann–Whitney test, p=$7.4 \times 10^{-9}$). Recombination rate does not differ significantly between old-X and neo-X (Mann–Whitney test, p=0.629).

(**Figure 17**). This increase of male-biased orphans among the older age classes is the result of a preferential loss of orphans with an unbiased gene expression (**Figure 18**). To confirm that male-biased gene expression is associated with orphan retention rather than emergence, we analyzed the sex-bias in *D. miranda* for orphans with and without an open reading frame. Consistent with the gene expression pattern in *D. pseudoobscura*, we found that lost orphans have a significantly lower male-bias in *D. miranda* (**Figure 19**). We conclude that the previously reported male-biased gene expression of orphan genes is not the result of a preferential recruitment of male-biased transcripts, nor do orphans gradually acquire male-biased gene expression. Rather, male-biased orphans are more likely to be retained.

## Discussion

Our study provides the missing link to understand orphan dynamics. Until now, orphan evolution was primarily studied on long phylogenetic branches. Although this approach is well suited to discover new orphans, it does not allow tracing the evolution of orphans. Previous studies showed a high rate of orphan gain, which is not reflected in an increase in gene number. To resolve this apparent paradox, it has been postulated that orphans must be lost at a high rate as well (**Tautz and Domazet-Loso, 2011**). In this study, we used the framework of closely related species in the *obscura* group to study the patterns of orphan gain and losses. We show that orphans not only emerge at high rates, but that they are also rapidly lost (**Figure 10**). Interestingly, most losses (~76%) were due to disabling mutations rather than deletions of the orphan gene. Although under equilibrium conditions the number of

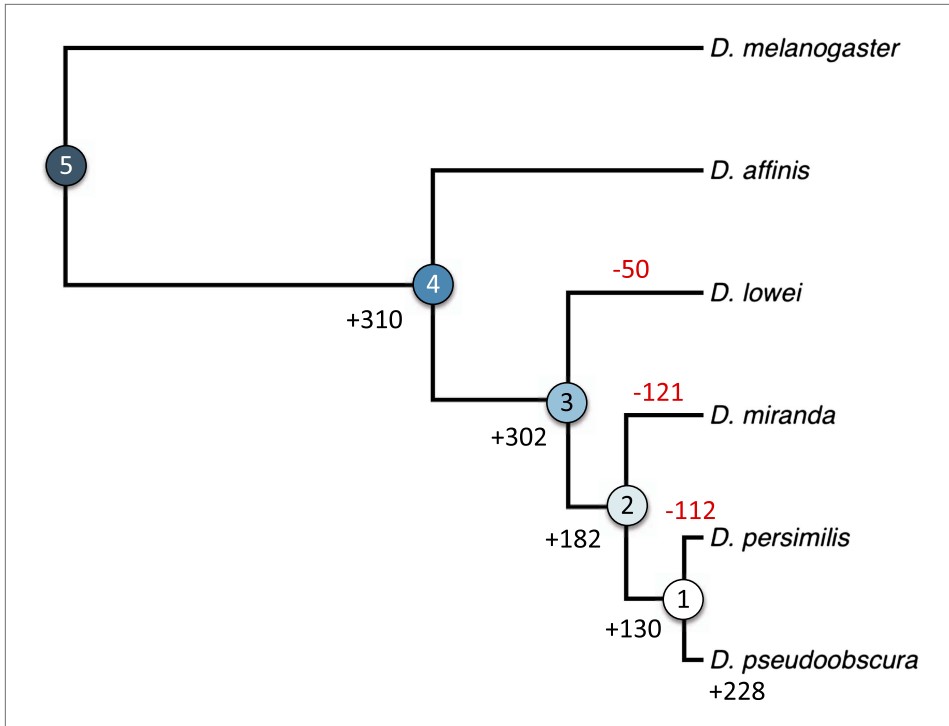

**Figure 6**. Orphan gain and losses in the *Drosophila obscura* group. Schematic phylogenetic tree of the *Drosophila obscura* group species according to ***Beckenbach et al. (1993)*** with *D. melanogaster* as outgroup. Genes conserved between *D. pseudoobscura* and 10 non-*obscura Drosophila* species correspond to age class 5 (old genes). For each age class the number of gene gains is shown in black. Orphans lost at a given branch are indicated in red. Note that losses at internal branches cannot be calculated, since all the orphans are present in *D. pseudoobscura*. Losses in *D. affinis* cannot be unambiguously assigned due to the absence of an additional *obscura* outgroup.

The following figure supplements are available for figure 6:

**Figure supplement 1**. Schematic tree of the Drosophila species analyzed in this study.

losses balances the number of orphan gains, here, we observed a surplus of orphan gains (***Figure 6***). We caution that this discrepancy probably does not imply an increase of gene number, but rather reflects the limited evolutionary time to acquire mutations. Using a rather conservative criterion for disabling mutations, either premature stop codons or frameshift indels, we have probably not identified all orphans that have lost their function. Furthermore, we do not account for the possibility of loss of function due to changes in gene regulation.

Importantly, codon usage bias, *dN/dS* values and sequence conservation clearly suggest that orphan genes are functionally constrained and these constraints do not differ among orphans that are conserved in the *obscura* group and those that lost function in at least one species of the group. Hence, it may be possible that orphan loss is stochastic and reflects weak purifying selection. Nevertheless, lost orphans differ in some aspects from conserved ones. Orphans that are lost contain more microsatellite stretches and have a lower, less sex-biased gene expression than retained ones. Furthermore, we also found that the rate of orphan loss decreases with orphan age, a result consistent with orphans serving a functional role only temporarily. Previous work suggested that orphans are important for adaptation to novel environments (***Khalturin et al., 2009***; ***Colbourne et al., 2011***), but it is also possible that orphans contribute to stabilize new connections in gene networks (***Capra et al., 2010***; ***Warnefors and Eyre-Walker, 2011***) and become obsolete once such new connections have been optimized. Our data suggest that orphans become quickly functional, which is reflected in their codon usage bias, *dN/dS* ratio and sequence conservation.

The chromosomal translocation resulting in the neo-X chromosome provides another interesting perspective on the evolution of orphan genes. Despite the fact that the neo-X is now fully dosage

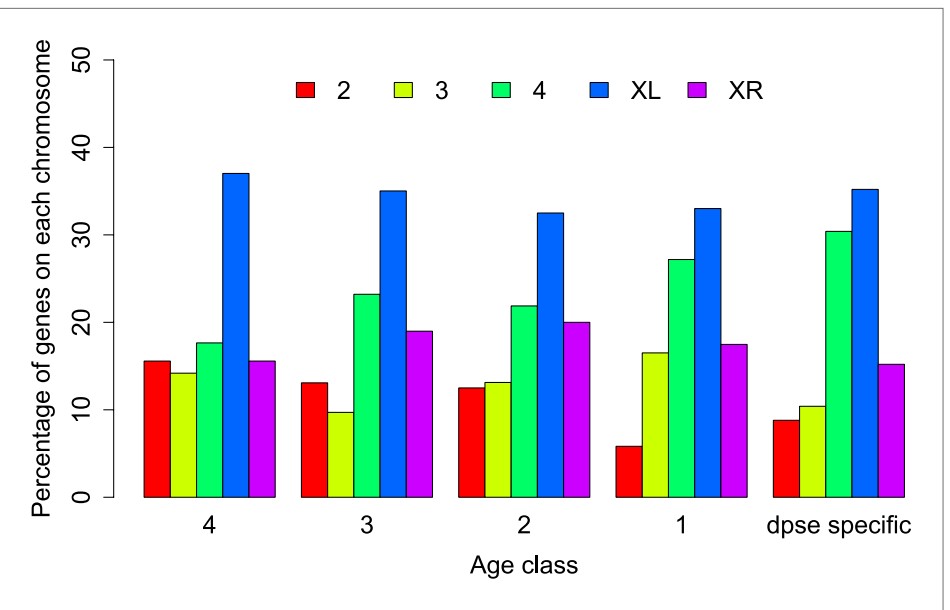

**Figure 7**. Chromosomal distribution of orphans of different age classes. In each age class orphans are underrepresented on the neo-X (XR) compared to old-X (XL) (Age class 4: χ²-test, p=6.3 × 10⁻⁹; age class 3: χ²-test, p=4.4 × 10⁻⁵; age class 2: χ²-test, p=0.00590; age class 1: χ²-test, p=0.00876; *D. pseudoobscura* specific: χ²-test, p=0.00030).

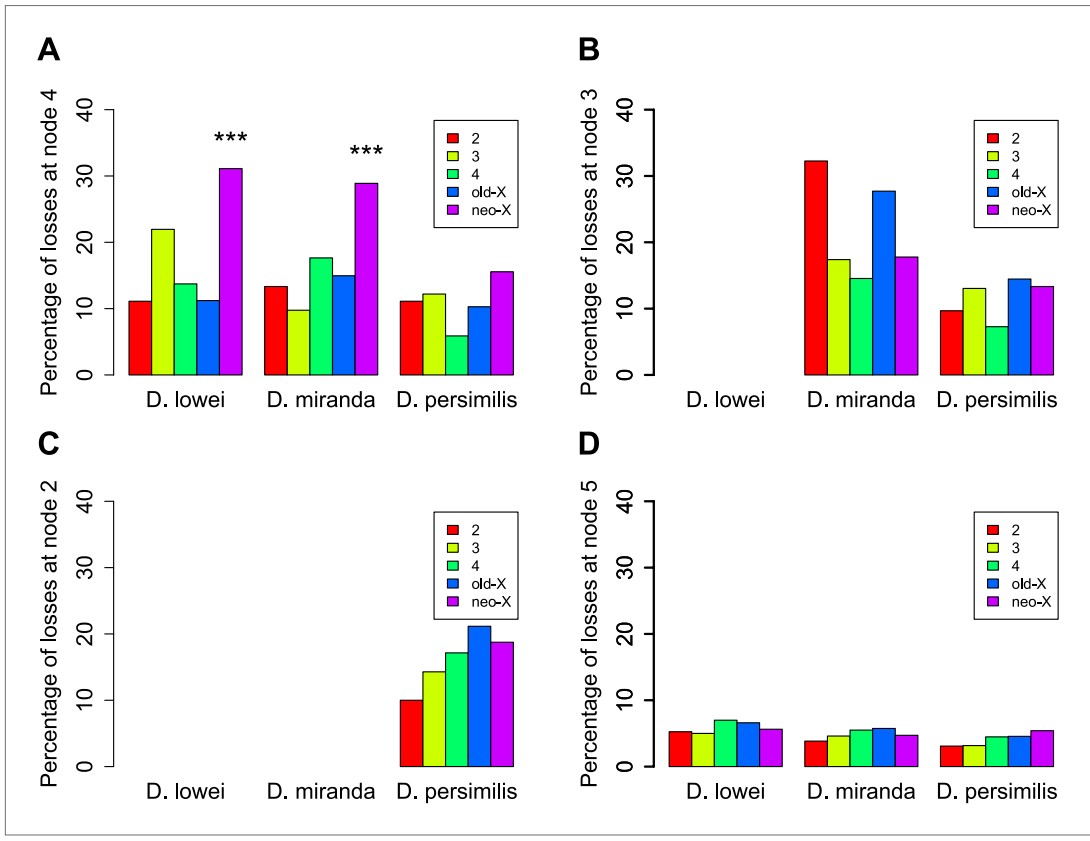

**Figure 8**. Orphans predating the XL-XR fusion are preferentially lost on the neo-X. For three terminal branches (*D. lowei*, *D. miranda*, and *D. persimilis*) the fraction of lost genes for each age class is shown. Each autosome and both X-chromosome arms are shown in different color. At node 4, where the neo-X originated, we observed the highest rate of orphan pseudogenization on the neo-X (**A**). Notably, this effect is not seen for younger orphans (**B** and **C**) neither for old genes (**D**).

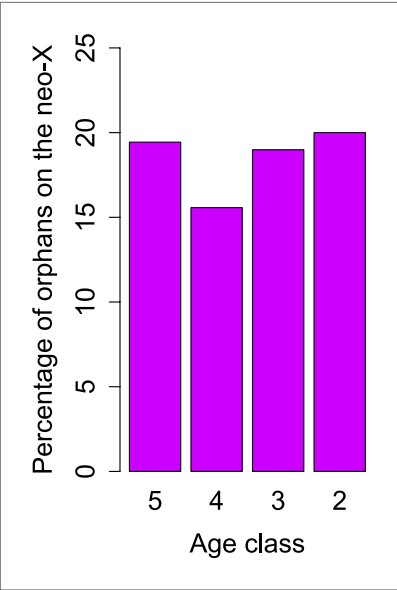

**Figure 9**. No change in orphan gain on the neo-X chromosome. The percentage of orphan genes on the neo-X chromosome remains constant through time (indicated by age classes).

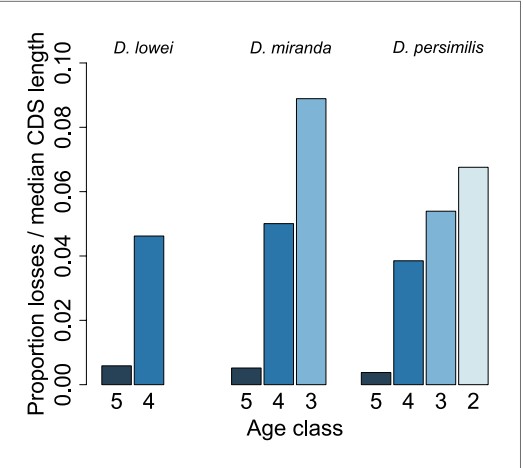

**Figure 11**. Young orphan genes are more likely to be lost: accounting for CDS length. To test if the short CDS of orphans affects the pattern that young orphans are more likely to lose function, we normalized the percentage of losses by the median CDS length of genes at that node.

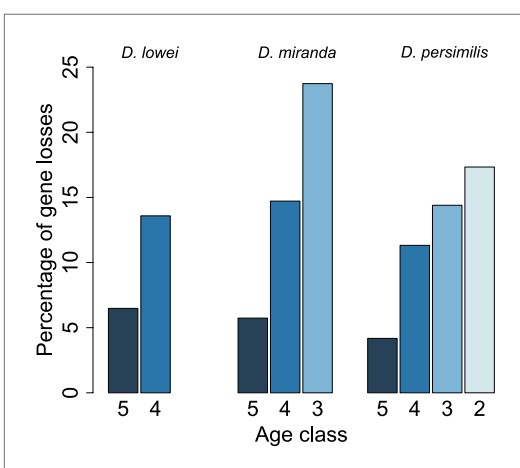

**Figure 10**. Young orphan genes are more likely to be lost. The barplot shows the fraction of orphans that has acquired a frameshift or premature stop codon (i.e., lost function). For *D. lowei*, *D. miranda*, and *D. persimilis*, the fraction of lost orphans is shown for different age classes. Orphans are more likely to be lost than old genes. Both the *D. miranda* and *D. persimilis* lineage show that younger orphans are more likely to lose function than older ones.

compensated (*Abraham and Lucchesi, 1974*), and has obtained a similar base composition as the XL (*Gallach et al., 2007*), we noted that the translocation resulted in a preferential loss of orphan genes on the neo-X. Since this pattern is restricted to orphans that most likely originated before the chromosomal fusion, we argue that the change in chromosomal environment has affected the function of orphan genes, most likely via expression differences. We speculate that the selective advantage conferred by these orphans has diminished, which resulted in a higher loss rate. Interestingly, the elevated rate of orphan loss after the neo-X formation seems to be still ongoing. This differential loss of orphan genes point in a similar direction as the observation that the gene composition of the neo-X has been altered by gene duplication (*Meisel et al., 2009*). Hence, both (orphan) gene loss and duplication contribute to fast gene content remodeling on a newly formed sex chromosome in *Drosophila*.

# Materials and methods

## Species data collection

An individual species sample of *D. affinis* (stock number 140120141.02) was ordered from the Drosophila Species Stock Center (https://stock-center.ucsd.edu/info/welcome.php) and sequenced on the Illumina GAIIx following the paired-end library preparation protocol (version Illumina 1.7) in two runs (run 1: read length = 101 bp, insert size = 230 bp; run 2: read length = 101 bp, insert size = 550 bp). Short genomic reads for *D. lowei* (accessions

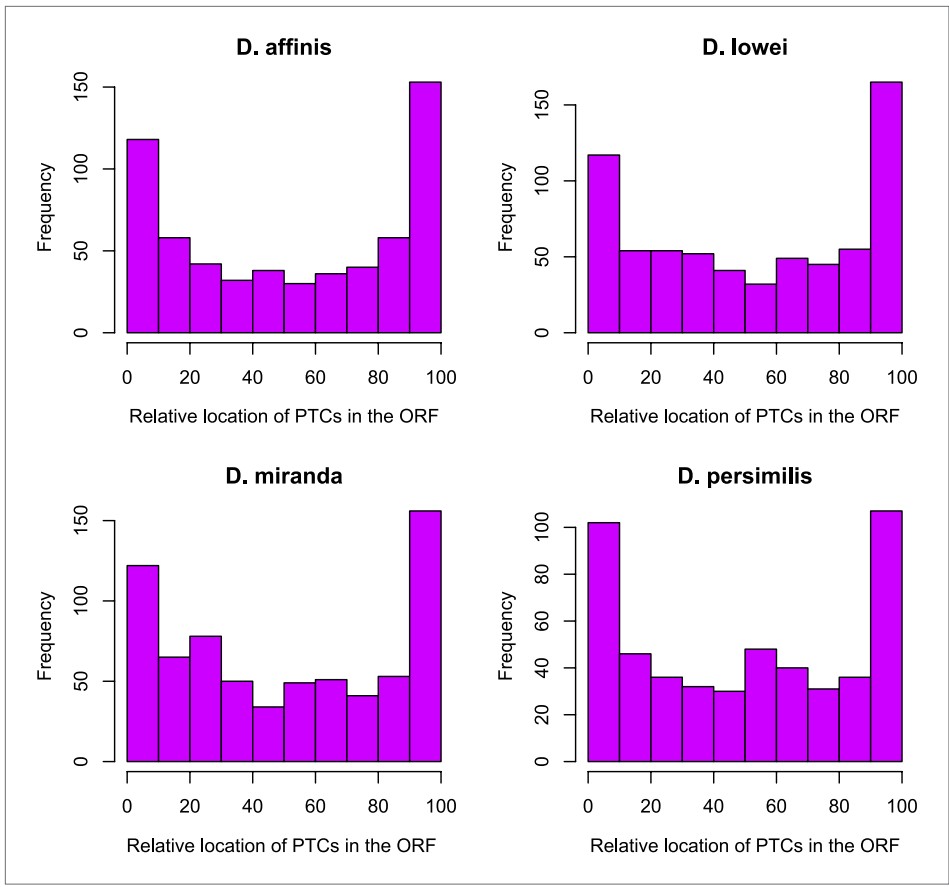

**Figure 12**. Distribution of premature stop codons (PTCs) along the ORF for all genes containing PTCs. PTCs are enriched at the beginning and at the end of the ORF in each species.

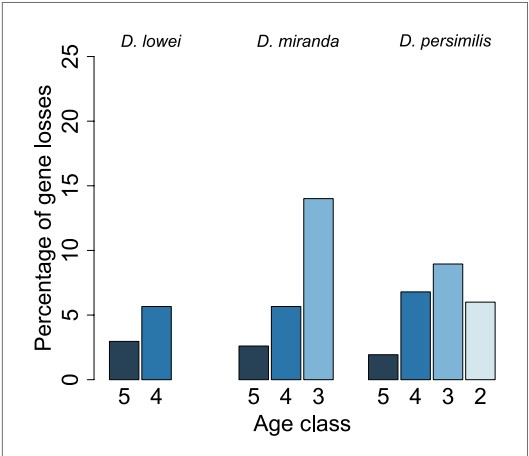

**Figure 13**. Young orphan genes are more likely to be lost: considering only frameshifts and premature stop codons occurring in the first half of the ORF. We repeated the analysis shown in *Figure 10* by considering only frameshifts and premature stop codons occurring in the first half of the ORF to define a conservative set of pseudogenes, since disrupting mutations occurring at the end of the ORF are likely to have little impact on the gene function.

SRX091466 and SRX091467) and *D. persimilis* (accession SRX091471) were downloaded from the Sequence Read Archive (http://www.ncbi.nlm.nih.gov/sra). The genome of *D. miranda* was downloaded from NCBI (GenBank Assembly ID GCA_000269505.1). The genome of *D. pseudoobscura* was downloaded from FlyBase (release 2.23).

## Assembly of the *obscura* species

Reads for *D. affinis*, *D. lowei*, and *D. persimilis* were trimmed using the Perl script trim_fastq.pl (parameters –quality-threshold 20 ––min-length 40) from PoPoolation (*Kofler et al., 2011*). For each species, a de novo assembly (parameters: min-contig-length 200) was performed using CLC Genomics Workbench 4.6 (http://www.clcbio.com/products/clc-genomics-workbench/), followed by scaffolding with nucmer (parameters: –c 30 –g 1000 –b 1000 –l 15) against the *D. pseudoobscura* genome. Average coverage per assembled genome was calculated by realigning the reads against the contigs of the respective

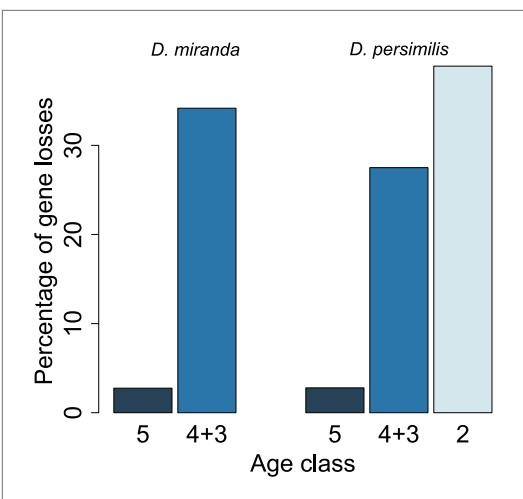

**Figure 14**. Young orphan genes are more likely to be lost: the conservative set of orthologs. We repeated the analysis shown in **Figure 10** by restricting it to orthologs for which at least one flanking gene is identified in the same contig (see 'Annotation of the *obscura* species'). Due to the substantially reduced number of orphans in the older age classes, we combined age class 3 and 4.

species with Bowtie 2.1.0 (parameters: --very-fast) and selecting only reads with mapping quality >20.

## Annotation of the *obscura* species

The annotation of *D. affinis*, *D. lowei*, *D. miranda*, and *D. persimilis* is based on orthology to *D. pseudoobscura* using Exonerate 2.2.0 (parameters: -model protein2genome–bestn 1 -show-targetgff), by aligning the longest isoform of *D. pseudoobscura* proteins extracted from a recent re-annotation of *D. pseudoobscura* (**Palmieri et al., 2012**) to the genomes of *D. affinis*, *D. lowei*, *D. miranda*, and *D. persimilis*. For each gene, the best unambiguous hit was retained. To remove non-informative hits, we also required a minimum fraction of the gene to be recovered. Since the sequence conservation of orthologs decreases with divergence time, the expected length of the ortholog depends strongly on the phylogenetic distance between query and subject sequence. To apply consistent criteria for all species, we empirically determined the expected fraction of a gene with sequence homology. Based on genes that are conserved between *D. pseudoobscura* and the 10 *Drosophila* species outside the *obscura* clade (old genes) (**Figure 6—figure supplement 1**), we determined the distribution of the fraction of the genes that could be aligned. As cutoff the value we used the 5[th] percentile of the distribution of aligned protein length of old genes. This resulted in a threshold of 47% for *D. affinis*, 52% for *D. lowei*, 59% for *D. miranda* and 53% for *D. persimilis*. Hence, only orphan orthologs that showed a fraction of aligned coding sequence higher than the empirically determined cutoffs were retained. In addition to this ortholog set, we generated an alternative, more conservative ortholog set. For this one, at least one of the flanking genes of *D. pseudoobscura* was required to be in synteny with the respective orthologs in *D. affinis*, *D. miranda* and *D. persimilis*. *D. lowei* was not considered in the synteny analysis since most of the genes in this species are flanked by genomic gaps, due to the shorter contig length of the *D. lowei* assembly (**Table 1**), which caused many contigs to contain only a single gene (**Table 2**), thus precluding proper synteny assignments. Assembly and annotation of all the species are available at http://popoolation.at/affinis_genome, http://popoolation.at/lowei_genome, http://popoolation.at/miranda_genome and http://popoolation.at/persimilis_genome. Detailed annotation statistics for each gene are available at 10.5061/dryad.hq564 (**Palmieri et al., 2014**).

## Detection of orphan genes

*D. pseudoobscura* proteins corresponding to the longest isoform for each gene were aligned using BLASTP (E < 10[−4]) and TBLASTN (E < 10[−4]) against the published proteomes and genomes of 10 *Drosophila* species outside the *obscura* group (*D. melanogaster*, *D. simulans*, *D. sechellia*, *D. erecta*, *D. yakuba*, *D. ananassae*, *D. willistoni*, *D. mojavensis*, *D. virilis*, *D. grimshawi*). Genes without BLAST hits and without annotated orthologs in FlyBase (gene orthologs release 09-2011) were classified as orphans.

## Polymorphism analysis

Illumina reads for 45 *D. pseudoobscura* strains were downloaded from NCBI (Sequence Read Archive, accession SRP017196). Reads were trimmed using PoPoolation (**Kofler et al., 2011**) and a total of 3.5 million reads was randomly extracted for each strain and combined into a single FASTQ file. The combined reads were treated as a Pool-Seq dataset and mapped to the FlyBase *D. pseudoobscura* genome release 2.23 with BWA (**Li and Durbin, 2009**) (parameters -o 1 -n 0.01 -l 200 -e 12 -d 12) on

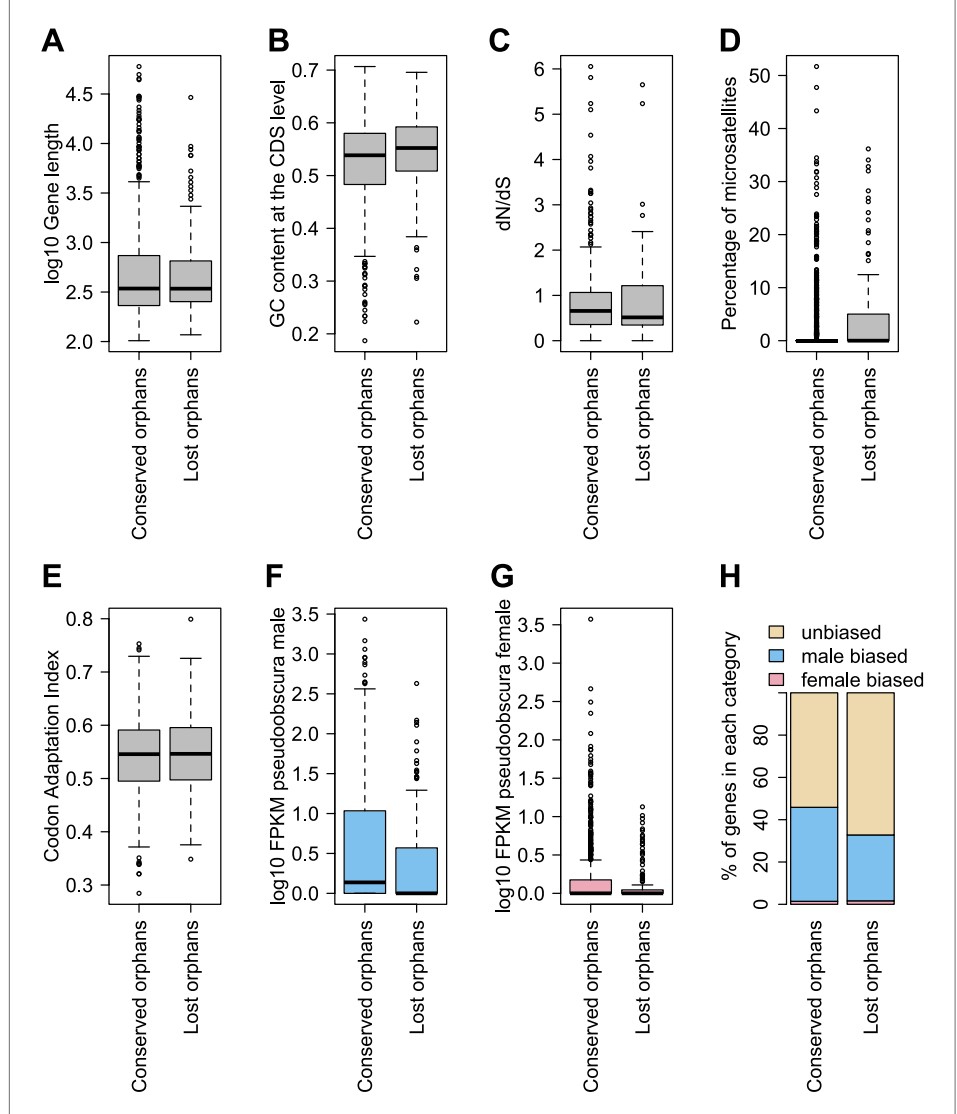

**Figure 15**. Features of conserved orphans vs lost orphans measured in *D. pseudoobscura*. (**A**) Gene length (**B**) GC content, (**C**) *dN/dS* (**D**) percentage of microsatellites in coding sequence (**E**) Codon Adaptation Index (**F**) gene expression levels in *D. pseudoobscura* males (**G**) gene expression levels in *D. pseudoobscura* females (**H**) sex-biased expression. Gene length (Mann–Whitney test, p=0.7235) and evolutionary rates (Mann–Whitney test, p=0.5835) are not significantly different between conserved and lost orphans. Lost orphans have higher GC content (Mann–Whitney test, p=0.00325), lower expression in *D. pseudoobscura* males (Mann–Whitney test, p=0.00012) and females (Mann–Whitney test, p=0.00230) and a higher microsatellite content (Mann–Whitney test, p=0.00049) compared to conserved orphans. Lost orphans are enriched in unbiased genes compared to conserved orphans ($\chi^2$-test, p=0.02611).

a hadoop cluster using DistMap (***Pandey and Schlötterer, 2013***). From the resulting BAM file, PCR duplicates were removed with Picard (http://picard.sourceforge.net) using the tool MarkDuplicates.jar (parameters REMOVE_DUPLICATES = true, VALIDATION_STRINGENCY = SILENT). Proper-pairs with mapping quality >20 were extracted with samtools (version 0.1.18) (***Li et al., 2009***). Indels were detected with PoPoolation using the script identify-genomic-indel-regions.pl (parameters--min-count 2 --indel-window 5) and masked from the reference genome prior to SNP calling. Coverage was subsampled to 50X for all the chromosomes. Only SNPs on the 3rd chromosome were considered in all analyses, since a balancer chromosome was used to extract the 3rd chromosome, precluding an

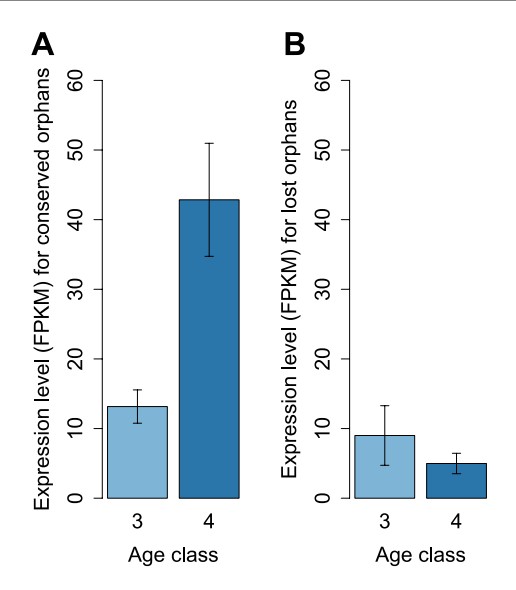

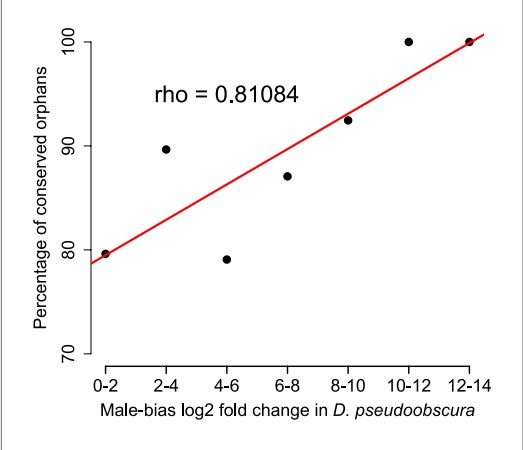

**Figure 16**. Conserved and lost orphans differ in their gene expression pattern. Expression intensity and sex bias in *D. miranda* for orphans conserved in all the *obscura* species (conserved orphans) vs orphans that pseudogenized in *D. lowei* and/or *D. persimilis* (lost orphans). Expression is calculated in males for orphans of age classes 3 and 4. Expression level increases with age for conserved orphans (**A**), while it decreases for lost orphans (**B**).

**Figure 18**. Conservation of orphans is correlated with male-biased gene expression. Orphans with male-biased gene expression in *D. pseudoobscura* were grouped into classes according to expression bias strength. The fraction of conserved orphans in each bin shows a significant positive correlation with expression bias (Spearman's *rho* = 0.811, p=0.02692). This correlation suggests that orphans with a more pronounced male-biased expression tend to persist longer than less male-biased orphans. No similar trend was seen for female-biased orphans (Spearman's *rho* = 0.78262, p=0.1176).

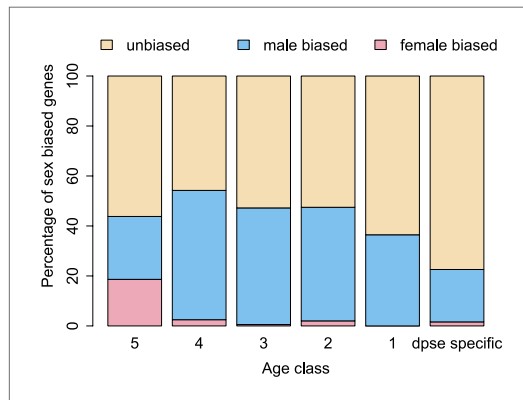

**Figure 17**. The proportion of male-biased orphans increases with age. Sex-biased expression was measured in *D. pseudoobscura* for orphans belonging to different age classes and for old genes (age class 5).

unbiased polymorphism analysis for the remaining chromosomes. SNPs were called with the PoPoolationscriptVariance-sliding.pl(parameters--min-coverage 10 --min-count 2 --max-coverage 500 --min-qual 20 --window-size 500 --step-size 500 --fastq-type sanger--pool-size 45).

## Calculation of orphan gains and losses

Orphan gains and losses (pseudogenizations) were inferred by Dollo parsimony. Based on the phylogenetic tree of *Figure 6*, a gene was assigned as gained at a given node if an intact ortholog was present in both external branches of the subtree corresponding to that node. For example, a gene having an intact ORF in *D. lowei* but not in *D. affinis* was classified as gained at node 3 (*Figure 6*). A gene was considered to be lost at a terminal branch if at least one ORF-disrupting mutation (frameshift/premature stop codon) was present in the gene at that branch and two intact ORFs were detected at both external leaves (*Wang et al., 2006*). The relatively high coverage of our assemblies (*Table 1*) makes unlikely that disrupting mutations are sequencing errors. In *D. affinis* for instance, only 8 genes had an average coverage lower than 20x.

A gene was considered as completely deleted in a species if no ortholog was detected in that species and no BLASTP (E < 10⁻⁴) or TBLASTN (E < 10⁻⁴) hit was found. Deletions were not

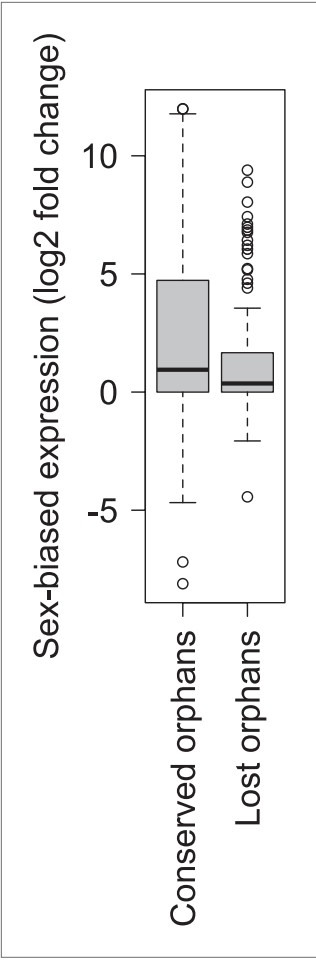

**Figure 19**. Comparison of strength of sex-biased gene expression for conserved and lost orphans in *D. miranda*. A sex-biased gene expression larger than zero indicates a higher gene expression intensity in males than in females (male-biased gene expression). Conserved orphans have significantly higher male-biased expression than lost orphans (Mann–Whitney test, p=0.03158).

considered into analyses of gene turnover, since they cannot be distinguished from missing annotations.

## Expression analysis

Four RNA-Seq datasets of *D. pseudoobscura* males and females (strains ps94 and ps88 from the ArrayExpress database—accession E-MTAB-1424), together with two RNA-Seq samples of *D. miranda* males and females from the Sequence Read Archive (accessions SRX106024, SRX106025), were used for expression analysis. For each sample, reads were trimmed using PoPoolation (*Kofler et al., 2011*) and aligned to the genome of the respective species with GSNAP version 2012-07-12 (*Wu and Nacu, 2010*) (parameters: –N 1). Only proper pairs mapping unambiguously to one position were retained. Expression in FPKM was calculated with Cufflinks version 1.2.1 (parameters: -F 0.10 –j 0.15 –I 300000). For *D. pseudoobscura* sex-bias was calculated using the package DESeq (*Anders and Huber, 2010*), treating the strains as two biological replicates for each sex and applying an FDR = 0.1. Differential expression between *D. miranda* males and females was calculated for both species using the $\log_2$ fold change on the normalized expression counts using the normalization protocol implemented in the R package DESeq (*Anders and Huber, 2010*) version 1.10.1.

## Codon usage bias

Codon usage bias was calculated using the R package seqinr (function cai) based on the *D. pseudoobscura* codon usage table downloaded from http://www.kazusa.or.jp/codon/cgi-bin/showcodon.cgi?species=7237.

## Evolutionary rates

Coding sequences of *D. pseudoobscura* and *D. miranda* orthologs without frameshifts/stop codons were aligned using PRANK (*Loytynoja and Goldman, 2005*) (parameters: –codon). To test for purifying selection on orphans, *dN/dS* was compared between orphans and a set of randomly selected intergenic regions. This set was generated as follows: (1) we identified the intergenic regions from the *D. pseudoobscura* annotation from *Palmieri et al. (2012)*, (2) for each CDS belonging to an orphan gene we extracted all the intergenic regions longer than that CDS, (3) we randomly selected one intergenic region and we extracted from that a random subregion with the same length of a given orphan CDS, (4) this procedure was repeated for all orphan CDS, resulting in a set of intergenic regions with the same length distribution as orphan CDSs. These regions were aligned with BLASTN (cutoff $10^{-5}$) to the *D. affinis* genome and for each region the best hit was kept and realigned with PRANK (default parameters) to the *D. pseudoobscura* query sequence. Each alignment was truncated at the 5'end to get an alignment length, which is a multiple of 3. Internal stop codons were replaced by Ns. The ratio of the rates of nonsynonymous and synonymous substitutions per gene (*dN/dS*) was measured using Markov models of codon evolution and maximum likelihood methods implemented in PAML (*Yang, 2007*).

**Table 1.** *De novo* assembly statistics

|  | *D. affinis* | *D. lowei* | *D. persimilis* |
|---|---|---|---|
| Number of contigs | 28,946 | 106,465 | 17,387 |
| N75 | 9,478 | 1,218 | 10,359 |
| N50 | 25,160 | 3,230 | 24,172 |
| N25 | 49,062 | 7,357 | 49,047 |
| Minimum length | 121 | 162 | 147 |
| Maximum length | 216,903 | 87,164 | 204,742 |
| Average length | 5,183 | 1,388 | 7,736 |
| Total bp | 150,030,247 | 147,756,871 | 134,501,523 |
| Average coverage | 51 X | 92 X | 44 X |

The *D. miranda* genome was available at NCBI, thus no de novo assembly was made for this species.

## Comparison of genomic features among old-X, neo-X, and autosomes

To shed light on the differences in orphan number between XL and XR, different features were compared among old-X, neo-X, and autosomes in *D. pseudoobscura* (unassembled contigs were not considered in this analysis): (A) GC content was calculated with the R package seqinr for 100 kb sliding windows along each chromosome (B) microsatellite density was calculated using SciRoKo 3.4 (*Kofler et al., 2007*) (parameters: -mode mmfp–l 15 –r 3 –s 15 –p 5 –seedl 8 –seedr 3 –mmao 3) for 100 kb sliding windows along each chromosome; (C) transposon density was estimated with RepeatMasker 3.2.9 (parameters: –q–gff -nolow–norna–species drosophila) for 100 kb sliding windows along each chromosome; (D) length of intergenic regions were calculated using BEDTools (-complement) by interval subtraction between genome and gene coordinates; (E) recombination rates for different windows were taken from *McGaugh et al. (2012)*.

### Microsatellite detection

Microsatellites were detected on the transcript sequences of the longest isoform for each *D. pseudoobscura* gene using the tool SciRoKo 3.4 (*Kofler et al., 2007*) (parameters: -mode mmfp–l 15 –r 3 –s 15 –p 5 –seedl 8 –seedr 3 –mmao 3).

### Transposons detection

Genomic annotation of transposons was performed in *D. pseudoobscura* using RepeatMasker 3.2.9 (parameters: –q–gff -nolow–norna–species drosophila). Only transposons longer than 50 bp and not overlapping with microsatellites (see 'Microsatellite detection') were retained. We required for an orphan to contain a full transposon sequence in one of its exons in order to classify it as associated with a transposon.

**Table 2.** Orthology annotation statistics

|  | *D. affinis* | *D. lowei* | *D. miranda* | *D. persimilis* |
|---|---|---|---|---|
| Total genes | 14,287 | 14,952 | 15,282 | 14,995 |
| Genes with frameshifts/PTC* | 1,233 | 1,266 | 1,171 | 898 |
| Mean number of genes per contig | 3.4 | 1.6 | – | 3.4 |
| Median number of genes per contig | 2 | 1 | – | 2 |
| Maximum number of genes per contig | 35 | 24 | – | 37 |

*PTC = Premature termination codons.

## Acknowledgements

We thank Ram Vinay Pandey for help with programming and Viola Nolte for support with genome annotations and library preparation. We are grateful to the members of the institute, in particular A Betancourt and R Kofler, for valuable discussion and comments on the manuscript.

## Additional information

### Funding

| Funder | Grant reference number | Author |
|---|---|---|
| Austrian Science Funds (FWF) | P22834 | Christian Schlötterer |

The funder had no role in study design, data collection and interpretation, or the decision to submit the work for publication.

### Author contributions

NP, Acquisition of data, Analysis and interpretation of data, Drafting or revising the article; CK, Analysis and interpretation of data, Drafting or revising the article; CS, Conception and design, Acquisition of data, Drafting or revising the article

## Additional files

### Major datasets

The following dataset was generated:

| Author(s) | Year | Dataset title | Dataset ID and/or URL | Database, license, and accessibility information |
|---|---|---|---|---|
| Palmieri N, Kosiol C, Schlötterer C | 2014 | Data from: The life cycle of *Drosophila* orphan genes | http://dx.doi.org/10.5061/dryad.hq564 | Available at Dryad Digital Repository. |

The following previously published datasets were used:

| Author(s) | Year | Dataset title | Dataset ID and/or URL | Database, license, and accessibility information |
|---|---|---|---|---|
| Zhou Q, Bachrog D | 2012 | *Drosophila miranda* Genome | http://www.ncbi.nlm.nih.gov/bioproject/PRJNA77213 | Publicly available at NCBI BioProject. |
| | | Short genomic reads *D. lowei* | http://www.ncbi.nlm.nih.gov/sra/SRX091466 | Publicly available at NCBI Sequence Read Archive (http://www.ncbi.nlm.nih.gov/sra). |
| | | Short genomic reads *D. lowei* | http://www.ncbi.nlm.nih.gov/sra/?term=SRX091467 | Publicly available at NCBI Sequence Read Archive (http://www.ncbi.nlm.nih.gov/sra). |
| | | Short genomic reads *D. persimilis* | http://www.ncbi.nlm.nih.gov/sra/?term=SRX091471 | Publicly available at NCBI Sequence Read Archive (http://www.ncbi.nlm.nih.gov/sra). |
| | | *D. pseudoobscura* genome | ftp://ftp.flybase.net/genomes/Drosophila_pseudoobscura/dpse_r2.23_FB2011_08/ | Publicly available at FlyBase (http://flybase.org). |
| | | Illumina reads for *D. pseudoobscura* strains | http://www.ncbi.nlm.nih.gov/sra/?term=SRP017196 | Publicly available at NCBI Sequence Read Archive (http://www.ncbi.nlm.nih.gov/sra). |

| | | |
|---|---|---|
| *D. miranda* MSH22 whole virgin male RNA-seq data | http://www.ncbi.nlm.nih.gov/sra/?term=SRX106024 | Publicly available at NCBI Sequence Read Archive (http://www.ncbi.nlm.nih.gov/sra). |
| *D. miranda* MSH22 whole virgin female RNA-seq data | http://www.ncbi.nlm.nih.gov/sra/?term=SRX106025 | Publicly available at NCBI Sequence Read Archive (http://www.ncbi.nlm.nih.gov/sra). |

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
