## [Decision Letter]

Thank you for sending your work entitled “The life cycle of *Drosophila* orphan genes” for consideration at *eLife*. Your article has been favorably evaluated by a Senior editor and 3 reviewers, one of whom is a member of our Board of Reviewing Editors.

The Reviewing editor and the other reviewers discussed their comments before we reached this decision, and the Reviewing editor has assembled the following comments to help you prepare a revised submission.

The authors discuss an interesting topic on life cycle of *Drosophila* orphan genes. They investigated the features of orphan genes in the *D. pseudoobscura* group based on an extensive new genomic dataset and by further analyses on substitution rate, gain/loss, and expression pattern of them they found that the orphan genes experienced purifying selection, and highly expressed orphan genes with strong male-bias are more likely to be retained. They proposed that orphan gene loss reflects lineage specific functional requirements.

The dataset consists of one newly sequenced genome as well as new assemblies of other genomes from read archives. The analysis is mostly solid and convincing. Overall, the paper represents a significant step forward in the field, but it requires further clarification of technical issues before it can be considered for acceptance in *eLife*.

The following comments are condensed from the comments of the three referees.

*Referee 1*:

My major comment concerning the methodology concerns the procedure to classify a “loss”. They say in the Methods section: “ Based on a set of orthologs in the obscura group, pseudogenes were identified by the presence of an intact ORF”, but Figure 9–figure supplement 2 shows that a large number of such disruptions occur in the last 10% of the reading frame (this is actually an interesting result that should not be relegated to the supplements). These might not be function-disabling mutations. I suggest to check relative conservation levels along the ORFs to assess this question or treat these genes differently in the overall comparisons.

My major comment concerning data analysis is towards the exclusive use of *D. miranda* for calculating divergence levels in Figure 1. Why is this not done for all species in the tree? And why *D. miranda*, rather than the newly sequenced *D. affinis* that is at the base of the comparisons?

All figures that support relevant data discussed in the text should be in the paper – supplementary figures should only be used for further clarifications.

*Referee 2*:

1) Loss of orphan genes is a key point of this study. To make the results convincing, calculation on gene gains and loses should be described in detail.

2) How exactly was the “intergenic” dataset constructed?

*Referee 3*:

1) When the average or median values are compared between gene classes, this comparison hides the potential fact that the distribution might include genes far from the mean. For example, even if Ka/Ks is on average <1 some orphan genes might be fast evolving and actually not easy to recognize in other species. These orphans are not “real” orphans but evolving to quick for the threshold of similarity to catch them. Authors should plot and compare the distributions not only means or medians.

In this sense, I do not think the authors have a conservative definition of orphans because they do not ask for Ka/Ks to be significantly smaller than 1. This would be conservative. Some might evolve as non-coding regions. Their definition is annotated genes of *D. pseudoobscura* that are expressed but show not similarity in other species at a particular threshold. However, whenever possible functionality should be supported with a Ka/Ks significantly <1 or Ka/Ks significantly >1 for the particular orphans to provide additional evidence of functionality. They should add a significance value in the supplementary table and comment on it in the text.

Is there any polymorphism data available for *D. pseudoobscura* to perform M-K test to test the functionality of the 225 new orphan genes in that genome?

2) Figure 5 should contain divergence times in the nodes. Figure 5 should be discussed in the light of the quality of every genome considered to make sure that losses or gains depend on the specific rates of orphan gene evolution per lineage and not on the quality of sequencing and assembly.

3) In the Results section I would like the authors to describe whether the loss of XR orphans is pseudogenization or actually a relocation. Can they comment if the gene has moved whenever possible to perform these analyses? They comment about the Meisel et al. work (2009). In that work some male-specific duplicates were found to relocate to autosomes. Is this the case for these orphans.

4) How were GAPs in the assembly treated? Can it be that some losses are actually GAPs in the assembly?

5) Can the high rate of orphans on the X chromosome be explained by the X being sequenced at a lower coverage if males and females were sequenced and this affecting the presence of genes in different assemblies? Please mention what individuals (males, females or 50%/50%) were sequenced for every genome.

6) In the Materials and methods section, an orphan is considered to be disabled if a single disrupting mutations is present. What is the probability that this single disabling mutation is actually a sequencing error? Please comment.

7) Intron turnover should be analyzed in more detail. How are those introns gained or lost? Are they part of the initially annotated coding region? This could reveal disablements. Some annotation software will put an intron in regions with disablements and still annotate a coding region.

8) In the Results section the authors mention that their definition of orphan is more conservative than in previous work. Please explain in detail in what sense the definition is conservative in this work.

9) Are orphans related (similar in sequence) to other orphans?

---

## [Author Response]

Referee 1:

*My major comment concerning the methodology concerns the procedure to classify a “loss”. They say in the Methods section: “ Based on a set of orthologs in the obscura group, pseudogenes were identified by the presence of an intact ORF”, but Figure 9–figure supplement 2 shows that a large number of such disruptions occur in the last 10% of the reading frame (this is actually an interesting result that should not be relegated to the supplements). These might not be function-disabling mutations. I suggest to check relative conservation levels along the ORFs to assess this question or treat these genes differently in the overall comparisons*.

We agree that disruptions present at the end of the ORFs might not be considered as real pseudogenizations events. To account for this, we looked at the correlation between gene age and loss by considering only the disrupting mutations occurring in the first half of the ORF. We obtained qualitatively similar results, as shown in Figure 13.

*My major comment concerning data analysis is towards the exclusive use of* D. miranda *for calculating divergence levels in*
Figure 1*. Why is this not done for all species in the tree? And why* D. miranda*, rather than the newly sequenced* D. affinis *that is at the base of the comparisons*?

We calculated now the divergence levels for all the species in the tree and obtained qualitatively similar results (Figure 1—figure supplement 2).

*All figures that support relevant data discussed in the text should be in the paper - supplementary figures should only be used for further clarifications*.

We moved all the figure supplements from Figure 9 (now Figure 10) to the main figures.

Referee 2:

*1) Loss of orphan genes is a key point of this study. To make the results convincing, calculation on gene gains and loses should be described in detail*.

We rewrote the methods section “Calculation of orphan gain and losses” providing a more detailed description.

*2) How exactly was the “intergenic” dataset constructed*?

We added the details in the method section “Evolutionary rates”.

Referee 3:

*1) When the average or median values are compared between gene classes, this comparison hides the potential fact that the distribution might include genes far from the mean. For example, even if Ka/Ks is on average <1 some orphan genes might be fast evolving and actually not easy to recognize in other species. These orphans are not “real” orphans but evolving to quick for the threshold of similarity to catch them. Authors should plot and compare the distributions not only means or medians*.

We plotted the distribution of dN/dS for orphans in Figure 1—figure supplement 1 and showed that most of the orphans have a dN/dS lower than 1.

*In this sense, I do not think the authors have a conservative definition of orphans because they do not ask for Ka/Ks to be significantly smaller than 1. This would be conservative. Some might evolve as non-coding regions. Their definition is annotated genes of* D. pseudoobscura *that are expressed but show not similarity in other species at a particular threshold. However, whenever possible functionality should be supported with a Ka/Ks significantly <1 or Ka/Ks significantly >1 for the particular orphans to provide additional evidence of functionality. They should add a significance value in the supplementary table and comment on it in the text*.

The reason why we called our orphan identification procedure conservative is that we included an additional filtering step that has not been used in previous publications. The procedure suggested by the reviewer is certainly a truly conservative one, but given the short evolutionary time scale of our study, too few mutations have occurred to identify genes with a statistically significant difference of Ka/Ks from one. We tested all orphans and none was found to have a Ka/Ks significantly different from 1. Hence, we did not further pursue the suggested strategy.

Since the reviewer feels that the wording “conservative” is a problem, we have removed it from the manuscript since we do not think that it is of major importance.

*Is there any polymorphism data available for* D. pseudoobscura *to perform M-K test to test the functionality of the 225 new orphan genes in that genome*?

We agree that the inclusion of polymorphism data could be interesting and we analyzed a set of polymorphism dataset for *D. pseudoobscura* from 45 individuals. Since the MK test is designed to identify genes with different patterns of evolution for divergence and polymorphism data, it is not well-suited for the identification of purifying selection. Hence, we used the polymorphism data set to calculate pN/pS for orphans (Figure 2) and compared them to conserved genes and intergenic regions. In support of our other data, also the polymorphism data supports the hypothesis of purifying selection acting on orphan genes.

*2)*
Figure 5
*should contain divergence times in the nodes.*
Figure 5
*should be discussed in the light of the quality of every genome considered to make sure that losses or gains depend on the specific rates of orphan gene evolution per lineage and not on the quality of sequencing and assembly*.

We added a new figure supplement (Figure 6—figure supplement 1), which includes the other Drosophila species and this figure includes scale with divergence times.

We understand the concern of the reviewer that the rates of gains and losses may also influenced by assembly quality, but we feel that this is difficult to put into this figure. Rather, we would like to refer the reviewer to Figure 10, which shows the percentage of lost non-orphan genes (age class 5). We caution, however, that it is not clear to what extent premature stops or frameshift mutations are segregating in these species or are the outcome of sequencing errors. We think that the high coverage of the genomes (Table 1) will result only in a small fraction of sequencing errors.

*3) In the Results*
*section I*
*would like the authors to describe whether the loss of XR orphans is pseudogenization or actually a relocation. Can they comment if the gene has moved whenever possible to perform these analyses? They comment about the Meisel et al. work (2009). In that work some male-specific duplicates were found to relocate to autosomes. Is this the case for these orphans*.

We performed this analysis by extracting the orphans pseudogenized on the XR in *D. miranda* and *D. lowei* and by aligning them to the respective genomes to find evidence for gene duplication. We found no support for this hypothesis, as described at in the Results section.

*4) How were GAPs in the assembly treated? Can it be that some losses are actually GAPs in the assembly*?

We understand the issue that GAPs in the assembly might confound the detection of genuine gene losses if the definition of loss is merely based on the presence/absence of the gene sequence in a given species. To avoid this problem we considered only pseudogenization events (i.e., premature stops or frameshift mutations), as specified in the methods. Thus GAPs in the assembly do not affect our definition of gene loss. At the same time, we also recognize that our definition of gene loss is an underestimate of the true number of losses.

*5) Can the high rate of orphans on the X chromosome be explained by the X being sequenced at a lower coverage if males and females were sequenced and this affecting the presence of genes in different assemblies? Please mention what individuals (males, females or 50%/50%) were sequenced for every genome*.

All the genome assemblies used in our study come from female individuals. This ensures similar coverage between autosomes and X chromosome. Thus, the high rate of orphans on the X cannot be explained by the X being sequenced at lower coverage. The strain details / accession numbers are given in the Methods. Nevertheless, we would point out that the coverage of the genome assemblies was high enough (Table 1), such that even a 50% lower coverage of males would not have resulted in a substantial increase in sequencing errors due to low coverage.

*6) In the Materials and methods section, an orphan is considered to be disabled if a single disrupting mutations is present. What is the probability that this single disabling mutation is actually a sequencing error? Please comment*.

Overall, our assemblies were not based on low coverage genomes (Table 1), thus sequencing errors are not very likely to contribute significantly to disrupting mutations, since this is expected for a coverage of 3 reads or less. However, we also think that a fraction of the disrupting mutations may still be segregating. We did a more careful analysis of disrupting mutations in *D. mauritiana* (a species for which we had assembled the genome ourselves and had also high quality polymorphism data available) and found consistent with other results in the literature that several of the interrupting mutations were not fixed in the population.

*7) Intron turnover should be analyzed in more detail. How are those introns gained or lost? Are they part of the initially annotated coding region? This could reveal disablements. Some annotation software will put an intron in regions with disablements and still annotate a coding region*.

The reviewer brings up a very important aspect. This made us recognize that intron turnover was not explored in sufficient detail in our study. Since we feel that this is not the main focus of our work and it would require in depth analyses, we removed the section about intron turnover.

*8) In the Results section the authors mention that their definition of orphan is more conservative than in previous work. Please explain in detail in what sense the definition is conservative in this work*.

Our definition of orphan is more conservative, since we used an additional filtering using TBLASTN. We explained how orphans were detected in previous studies at the beginning of the result section and how our filtering makes our definition more conservative. In any case, we do not consider this an essential aspect of the manuscript and have removed the statement that our definition of orphans is conservative.

*9) Are orphans related (similar in sequence) to other orphans*?

We clustered the protein sequences of orphan genes using the CD-hit software, widely used for gene family analyses. Generally, we find that only about 7% of the orphans show similarity to other orphans. We summarized the results of the orphan gene family analysis in the revised manuscript.